Research

 

**Subject Area:**
molecular biology

O-GlcNAcylation, CRMP2, crosstalk, cognitive function

**Author for correspondence:**
Daan M. F. van Aalten
e-mail: dmfvanaalten@dundee.ac.uk

# Loss of CRMP2 O-GlcNAcylation leads to reduced novel object recognition performance in mice

Villo Muha[1], Ritchie Williamson[1,3], Rachel Hills[4], Alison D. McNeilly[5], Thomas G. McWilliams[7], Jana Alonso[1], Marianne Schimpl[1], Aneika C. Leney[8,9,10], Albert J. R. Heck[9,10], Calum Sutherland[6], Kevin D. Read[2], Rory J. McCrimmon[5], Simon P. Brooks[4] and Daan M. F. van Aalten[1]

[1]Gene Regulation and Expression, and [2]Wellcome Centre for Anti-Infectives Research, School of Life Sciences, University of Dundee, Dundee DD1 5EH, UK
[3]School of Pharmacy and Medical Sciences, Faculty of Life Sciences, University of Bradford, Bradford BD7 1DP, UK
[4]Division of Neuroscience, School of Bioscience, Cardiff University, Cardiff CF10 3AX, UK
[5]Systems Medicine, and [6]Cellular Medicine, School of Medicine, University of Dundee, Dundee DD1 9SY, UK
[7]Stem Cells and Metabolism, Research Programs Unit, Faculty of Medicine, University of Helsinki, PL 63 Haartmaninkatu 8, Helsinki 00014, Finland
[8]School of Biosciences, University of Birmingham, Birmingham B15 2TT, UK
[9]Biomolecular Mass Spectrometry and Proteomics, Bijvoet Center for Biomolecular Research and Utrecht Institute for Pharmaceutical Sciences, University of Utrecht, Padualaan 8, 3584 CH Utrecht, The Netherlands
[10]Netherlands Proteomics Centre, Padualaan 8, 3584 CH Utrecht, The Netherlands

DMFvA, 0000-0002-1499-6908

O-GlcNAcylation is an abundant post-translational modification in the nervous system, linked to both neurodevelopmental and neurodegenerative disease. However, the mechanistic links between these phenotypes and site-specific O-GlcNAcylation remain largely unexplored. Here, we show that Ser517 O-GlcNAcylation of the microtubule-binding protein Collapsin Response Mediator Protein-2 (CRMP2) increases with age. By generating and characterizing a $Crmp2^{S517A}$ knock-in mouse model, we demonstrate that loss of O-GlcNAcylation leads to a small decrease in body weight and mild memory impairment, suggesting that Ser517 O-GlcNAcylation has a small but detectable impact on mouse physiology and cognitive function.

## 1. Introduction

Intracellular homeostasis and rapid cellular responses to extracellular stimuli are coordinated by combinations of different post-translational modifications on proteins. O-GlcNAcylation involves the addition of O-linked *N*-acetylglucosamine (O-GlcNAc) to serine/threonine residues on nucleocytoplasmic proteins [1–3]. Levels of protein O-GlcNAcylation are linked to levels of the sugar donor UDP-GlcNAc synthesized via the hexosamine biosynthetic pathway (HBP) that integrates input from carbohydrate, amino acid, fat and nucleotide metabolism [4]. O-GlcNAcylation has been proposed to fine-tune protein function according to nutrient levels in conjunction with stress signals [5]. As a consequence of perturbations in cellular UDP-GlcNAc levels due to defects of metabolic homeostasis, abnormal levels of O-GlcNAcylation have been suggested to contribute to the development of chronic diseases such as diabetes, cancer and neurodegeneration [6,7].

Only two enzymes, O-GlcNAc transferase (OGT) and O-GlcNAc hydrolase (OGA), are responsible for establishing the dynamic O-GlcNAcome, by addition and removal of the modification, respectively [8–10]. It is well established that OGT is essential for mammalian embryogenesis, and $Ogt^{-/-}$ null mice are not viable [11–14]. OGA, encoded by a single gene (*Mgea5/Oga*), is also indispensable for mammalian development. Mice homozygous for $Oga^{-/-}$ mutation do

not survive beyond perinatal development and show defects in glycogen mobilization [15,16]. O-GlcNAcylation is particularly important in the nervous system. Neuron-specific genetic ablation of *Ogt* in mice results in severely attenuated neurodevelopment [17]. Furthermore, loss of *Ogt* in the adult mouse brain leads to neurodegenerative phenotypes [18]. Studies using conditional *Ogt* knock-out mice have revealed essential roles for O-GlcNAcylation in controlling appetite [19], browning of white adipose tissue through regulating Agouti-related protein neurons [20] and excitatory synapse maturation [21]. In humans, missense mutations in *Ogt* have recently been linked to the X-linked intellectual disability syndrome OGT-XLID [22–26].

Despite the identification of numerous O-GlcNAc modification sites in over 3000 proteins, little is known about their physiological and functional significance *in vivo*. It also remains unknown how specific modifications could contribute to the severe phenotypes observed in $Ogt^{-/-}$ and $Oga^{-/-}$ null models, as well as human OGT-XLID or chronic diseases. O-GlcNAcylation has been implicated in a large spectrum of cellular processes [27,28], including transcriptional regulation [29], signal transduction networks [30,31], protein folding [32], mitochondrial function [33,34] and protein degradation [35]. Driven by converging pre-clinical and pathological insights associated with loss of OGT function, we sought to identify candidate O-GlcNAc proteins underlying these phenotypes.

Proteomics studies have suggested the presence of O-GlcNAc on Collapsin Response Mediator Protein-2 (CRMP2), one of the most abundant neuronal proteins that binds to tubulin heterodimers and promotes microtubule assembly [36]. The C-terminal disordered region of CRMP2 is O-GlcNAcylated at a single position, within a region that harbours key CDK5/GSK3β regulatory phosphosites [37,38]. These sites are known to be targeted by axon-guiding Semaphorin3A/PlexinA signalling [39]. O-GlcNAcylation has been proposed to counteract hyperphosphorylation of Tau, possibly opposing the formation or propagation of pathogenic neurofibrillary tangles associated with Alzheimer's disease (AD) [40]. CRMP2 hyperphosphorylation has been observed in neurofibrillary tangles of AD patient brain tissue [41]. Furthermore, CRMP2 hyperphosphorylation is an early phenotypic event in pre-clinical mouse models of AD, occurring prior to the onset of inclusion pathology [41,42]. Elevated levels of phospho-CRMP2 have also been identified in breast cancer [43] and non-small cell lung cancer (NSCLC) [44]. Oncogenic potential is regulated by phosphorylation of the nuclear isoform, CRMP2A, at Ser522 [45], highlighting the importance of phospho-CRMP2 in chronic disease states. Given the position of the CRMP2 O-GlcNAc site, it is plausible that there is interplay with this regulatory phosphorylation, as has been proposed for other proteins [46–49].

Under normal conditions, CRMP2 controls cellular processes involving active rearrangements of microtubules such as neurite outgrowth, centrosome positioning and motility [50]. CRMP2 (encoded by *DPYSL2*) has also been implicated in anterograde axonal transport of various neuronal proteins such as TrkB [51] and the Sra1/WAVE1 complex [52]. Furthermore, CRMP2 interacts with the $Ca^{2+}$-binding protein CaM, the N-type voltage-gated calcium channel (CaV2.2) and the NMDA receptor (NMDAR) subunits NR2A/2B [53]. Taken together, CRMP2 serves as an important adaptor/scaffold protein central to neuronal function [53]. Although

viable, $Crmp2^{-/-}$ mice exhibit aberrant dendritic and synaptic development, leading to abnormal locomotion and social behaviour [54–56]. The CRMP2–tubulin interaction is regulated by Cdk5, which phosphorylates CRMP2 at Ser522 [57], allowing for subsequent processive phosphorylation of CRMP2 at Thr509, Thr514 and Ser518 by GSK3β [57,58]. This multi-site phosphorylation restricts the ability of CRMP2 to interact with tubulin, leading to growth cone collapse and neurite retraction [57]. Previous work has demonstrated that O-GlcNAcylation blocks hyperphosphorylation on a peptide derived from the corresponding C-terminal tail sequence *in vitro* [38]. Conversely, Thr514 phosphorylation hampers O-GlcNAcylation, thus suggesting a possible regulatory role for the Ser517 O-GlcNAc site.

Here, we demonstrate that the sole O-GlcNAcylation site on CRMP2 *in vivo* is located at Ser517, within the flexible C-terminal tail of the protein that is unique among the CRMP family. We reveal that O-GlcNAcylation at Ser517 is inducible and dynamic, increasing with age in the human brain. To dissect the physiological role of CRMP2 O-GlcNAcylation, we generated $Crmp2^{S517A}$ knock-in mice, which exhibited significant effects on body weight and cognitive function. Our study highlights the physiological importance of single O-GlcNAc modifications, establishing a framework for similar discoveries in the wider O-GlcNAc proteome.

# 2. Results and discussion

## 2.1. CRMP2 is O-GlcNAcylated at Ser517 *in vivo*

Several neuronal-specific proteins have been found to be modified by O-GlcNAc at serine and threonine residues, including CRMP2, an abundant protein involved in axonal guidance [59–61]. *In vitro* findings have suggested that CRMP2 is O-GlcNAcylated on Ser517 [38]. Remarkably, CRMP2 sequences including the O-GlcNAc site are highly conserved across mammalian species. CRMP2 is 99% identical in sequence in human, mouse, rat and sheep CRMP2. To investigate the sites of O-GlcNAcylation on CRMP2 *in vivo*, full-length CRMP2 was purified from sheep brain using ion exchange chromatography. Direct site mapping was not feasible, possibly because of low O-GlcNAcylation stoichiometry of CRMP2 in biological samples. However, we have exploited a recently reported O-GlcNAc protein enrichment method using the non-selective and high-affinity binding by a mutant version of the bacterial O-GlcNAcase from *Clostridium perfringens* ($CpOGA^{D298N}$) [62,63]. Following this approach, we performed site mapping of O-GlcNAc by mass spectrometry, leading to the identification of a single O-GlcNAcylation site at Ser517 (electronic supplementary material, figure S1), which is in agreement with previous reports [38]. The Ser517 O-GlcNAc site on mammalian CRMP2 is unique among members of the CRMP family (figure 1*a*) and the proximal amino acid sequence matches the peptide sequon for OGT recognition [61,64,65]. Thus, Ser517 appears to be the bona fide site for CRMP2 O-GlcNAcylation *in vivo*.

## 2.2. CRMP2 O-GlcNAcylation increases during human ageing

Overall levels of protein O-GlcNAcylation fluctuate during development and lifespan in the mammalian brain [15,66].

royalsocietypublishing.org/journal/rsob     Open Biol. 9: 190192

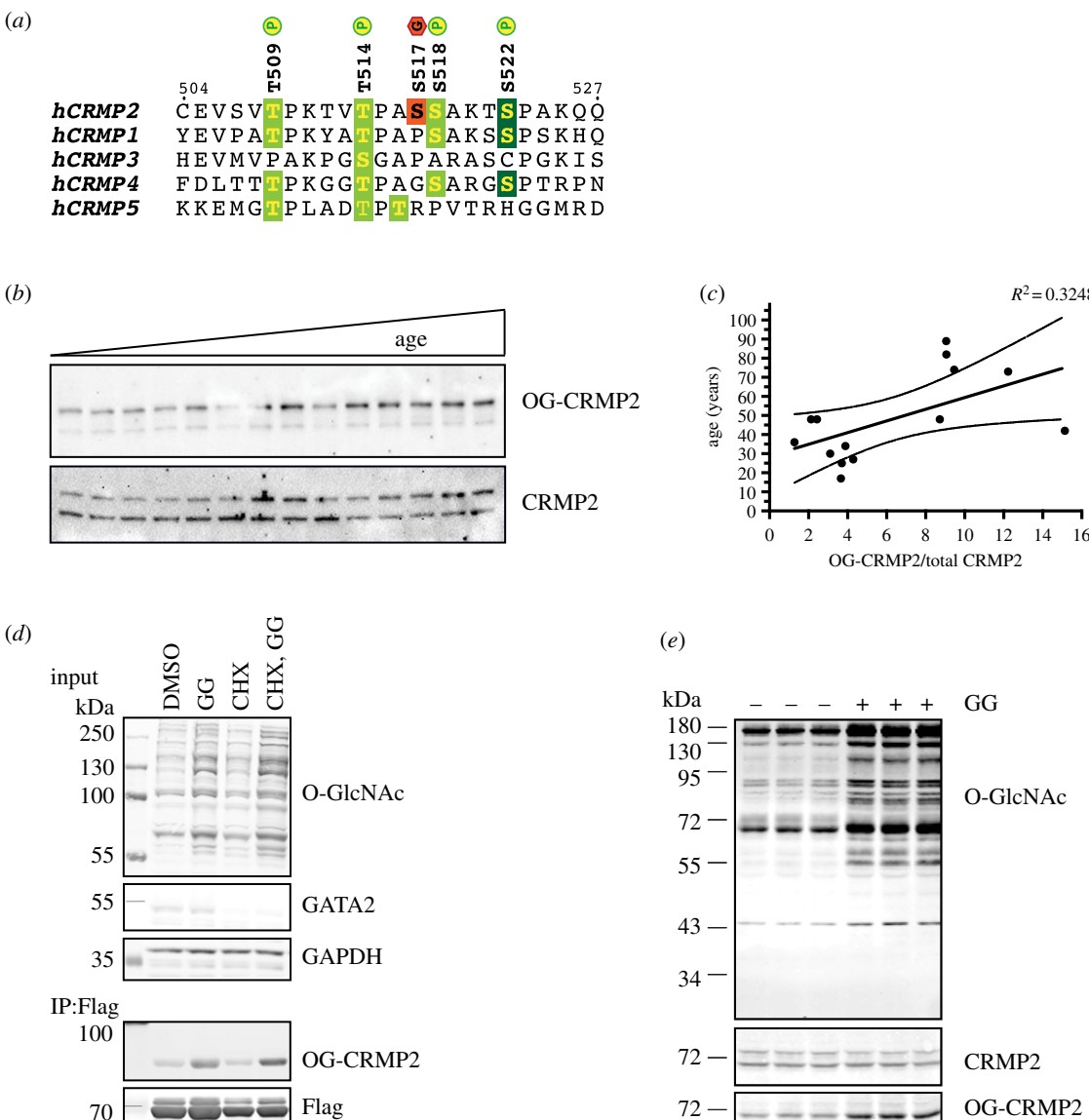

**Figure 1.** CRMP2 undergoes dynamic O-GlcNAc post-translational modification at Ser517 *in vivo*. (*a*) Sequence alignment shows that the O-GlcNAcylation modification site on the C-terminal tail is unique to CRMP2 among other members of the CRMP family. The Ser517 O-GlcNAcylation site is in close proximity to several phosphorylation sites targeted by Cdk5 (dark green) and GSK3β (light green) kinases. (*b*) Western blotting for O-GlcNAc-CRMP2 (OG-CRMP2) on post-mortem human hippocampus samples ($n = 14$, age range 17–89 years). (*c*) Correlation analysis reveals that the Pearson correlation coefficient for the OG-CRMP2 level versus age is 0.57, indicating a positive correlation. (*d*) O-GlcNAcylation of CRMP2 is post-translational. Neuro2A cells expressing Flag-tagged CRMP2 were treated with cyclo-heximide (CHX), a protein translation inhibitor, and subsequently supplemented with GlcNAcstatin G (GG) OGA inhibitor. Blocking translation did decrease CRMP2 protein levels, but it did not inhibit induction of O-GlcNAcylation on CRMP2. (*e*) O-GlcNAcylation of CRMP2 is inducible *in vivo*. Mice were infused (osmotic pump implant) with 10 mg ml$^{-1}$ GG ($n = 3$) or dose vehicle alone ($n = 3$) at 8 µl h$^{-1}$ for 24 h. Brain lysates were probed with site-specific OG-CRMP2 antibody on western blot. GAPDH, glyceraldehyde 3-phosphate dehydrogenase.

Although overall protein O-GlcNAcylation tends to increase with age in rodents, our temporal knowledge of site-specific O-GlcNAcylation events in mammalian tissues has remained largely obscure. To define and clarify the dynamics of a single O-GlcNAcylation modification *in vivo*, we generated anti-CRMP2 Ser517 O-GlcNAc site-specific antibody for this purpose (electronic supplementary material, figure S2). To investigate this selective O-GlcNAcylation event in a more human context, we profiled CRMP2 O-GlcNAcylation in healthy ageing and neurodegeneration using human post-mortem brain samples (electronic supplementary material, figure S2). Experiments with this antibody revealed that O-GlcNAcylation on CRMP2 gradually increases during the ageing process (figure 1*b*,*c*; Pearson correlation coefficient 0.57). We also found an increased trend of CRMP2 O-

GlcNAcylation in post-mortem brain samples from AD patients compared with aged-matched controls; however, this did not reach significance in the samples tested (electronic supplementary material, figure S3). Taken together, these data show that CRMP2 O-GlcNAcylation increases as a function of the ageing process in humans.

## 2.3. Site-specific O-GlcNAcylation of CRMP2 is post-translational and dynamic

O-GlcNAc glycosylation is a reversible and inducible post-translational modification [27,67–69]. For a limited subset of proteins, O-GlcNAcylation has been suggested to be co-translational, preventing premature degradation of nascent

polypeptides [70]. We next assessed the post-translational or co-translational nature of O-GlcNAcylation on CRMP2, by probing cycling of the O-GlcNAc site using chemical inhibition of protein synthesis. We treated murine neuroblastoma Neuro-2a cells expressing Flag-CRMP2 with cycloheximide and subsequently with GlcNAcstatin G, a highly selective OGA inhibitor that enhances cellular O-GlcNAc levels [71,72]. Although blocking protein translation diminished CRMP2 protein levels, it did not alter induction of O-GlcNAcylation on CRMP2 (figure 1d).

We next sought to determine the dynamic nature of this modification *in vivo*. The most potent and selective OGA inhibitor currently available is GlcNAcstatin G [72]. This molecule is able to raise O-GlcNAc levels in cells by disrupting the balance between O-GlcNAc transfer and hydrolysis, yet it has not been explored whether this enhancement can occur in the context of a whole organism. We infused ($8\,\mu l\,h^{-1}$) female NMRI mice ($n = 7$) over a 24 h period with GlcNAcstatin G ($n = 4$, $10\,mg\,ml^{-1}$) or dose vehicle only ($n = 3$) using Alzet osmotic pumps implanted subcutaneously with jugular vein cannulation. Pharmacokinetics was assessed in one mouse, with blood samples taken at 15, 30 min, 1, 2, 4, 6 and 21 h, then terminal blood and brain collected at 24 h. At 24 h, a terminal blood sample was also taken from all remaining mice ($n = 6$), and brains were harvested for biochemical analysis. GlcNAcstatin G concentrations in blood and brain were determined by ultra-performance liquid chromatography–tandem mass spectrometry (UPLC-MS/MS) following a suitable extraction procedure (electronic supplementary material, table S1). In the single animal assessed for pharmacokinetics, the brain : blood ratio was 0.07 at 24 h, with a brain concentration of GlcNAcstatin G of 99 ng ml$^{-1}$ (258 nM) (electronic supplementary material, table S1). Comparatively, the half-maximal effective concentration for GlcNAcstatin G determined in cultured HEK293 cells is 20 nM [72].

Brain lysates were probed with the pan-specific O-GlcNAc antibody RL2, revealing increased levels of protein O-GlcNAcylation, confirming inhibitor delivery to the brain (figure 1e). Immunoblotting using our site-specific Ser517 O-GlcNAc antibody (electronic supplementary material, figure S2) revealed elevated levels of this modification (figure 1e). These data provide evidence that site-specific O-GlcNAcylation of CRMP2 is both post-translational and dynamic.

## 2.4. CRMP2 phosphorylation by GSK3β is unchanged in Crmp2$^{S517A}$ mice

We next sought to clarify the physiological significance of CRMP2 O-GlcNAcylation in the vertebrate nervous system. To this effect, we generated a constitutive knock-in mouse model expressing endogenous CRMP2 that cannot be O-GlcNAcylated at Ser517 (Crmp2$^{S517A}$) (electronic supplementary material, figure S4). Homozygous Crmp2$^{S517A/S517A}$ mice are viable, survive to adulthood and appear healthy, showing no gross differences in physical characteristics from wild-type animals. Consistent with our targeting strategy, western blot analysis confirmed that endogenous CRMP2 O-GlcNAcylation on Ser517 was abolished in mouse brain lysate (figure 2a and electronic supplementary material, figure S2B). Furthermore, mutant CRMP2 Ser517Ala protein is expressed at a similar level to the wild-type CRMP2 (figure 2b,c), indicating the mutation does not influence protein stability.

Driven by the hypothesis that O-GlcNAcylation obscures phosphorylation on overlapping phosphosite motifs (figure 1a), we assayed the phosphorylation status of endogenous CRMP2 in wild-type and mutant brain lysates using mass spectrometry. CRMP2 protein was purified via a two-step ion exchange chromatography protocol from wild-type and Crmp2$^{S517A}$ mouse brain lysate, digested and phosphopeptides enriched prior to LC-MS/MS analysis. Samples were isolated from two independent mouse cohorts: young mice aged 45–46 days (wild-type $n = 4$ and Crmp2$^{S517A}$ $n = 4$) and a mature adult group aged 178–182 days (wild-type $n = 3$ and Crmp2$^{S517A}$ $n = 2$). We detected phosphorylated peptides in both groups, with phosphorylation on Ser522 observed in young mice and on Thr509, Thr514, Ser518 and Ser522 in samples from mature adult animals. Since more single and multi-phosphorylated peptides were detected on CRMP2 in the mature adults than in young mice (electronic supplementary material, table S2) we further profiled the phosphorylation status of CRMP2 in six-month-old mice. As previous *in vitro* studies indicated that GSK3β-mediated phosphorylation of Thr514 is influenced by O-GlcNAcylation [38], we employed antibodies that recognize the triphospho- (P509/514/522, P509/514 antibody) or tetraphospho- (P509/514/518/522, 3F4 antibody) forms of CRMP2. Contrary to our hypothesis, there was no detectable change in polyphosphorylation on CRMP2 at sites for GSK3β kinase (figure 2c–f). Our findings suggest that CRMP2 phosphorylation by GSK3β is unchanged in Crmp2$^{S517A}$ mice.

## 2.5. Loss of CRMP2 O-GlcNAcylation affects body weight

Animal weight is a well-established indicator of homeostasis and can reliably predict abnormal development, metabolism and neurological function. Decreased body weight was observed in conditional knock-out Crmp2$^{-/-}$ mice [56], yet this was not phenocopied in an independent Crmp2$^{-/-}$ knock-out study [54]. As part of our overall characterization, we monitored longitudinal body weight in a cohort of gender-matched, littermate wild-type and Crmp2$^{S517A}$ animals from mixed 50–50% C57BL/6NTac and C57BL/6 J genetic backgrounds. Interestingly, we observed a consistent incidence of decreased body weight in Crmp2$^{S517A}$ mutants (repeated measures of ANOVA, $p = 0.005$) compared with their gender-matched siblings (figure 3a,b). Crmp2$^{S517A}$ homozygous mice exhibited on average a 3–5% reduction in weight compared with their wild-type siblings at 12, 16, 20 and 24 weeks old (paired Student's $t$-test, parametric two-tailed $p$-value $= 0.0007$, $n = 13$, at 24 weeks of age) (figure 3b; electronic supplementary material, table S3). In a separate experiment, body weight measurements were repeated on male animals only, prior to behavioural characterization. For these tests we used a littermate mouse cohort from a mixed genetic background with a higher proportion of C57BL/6 J (12.5% C57BL/6NTac and 87.5% C57BL/6 J). In this group, body weight was not significantly different between wild-type and homozygous knock-in animals at age 15 weeks (electronic supplementary material, figure S5). We anticipated that the metabolic characteristics of the genetic backgrounds of the mice probably influenced the contribution of CRMP2 signalling to overall body weight [73]. Our findings indicate that loss of CRMP2 O-GlcNAcylation potentially influences the regulation of mammalian body weight in a context-dependent fashion.

royalsocietypublishing.org/journal/rsob   Open Biol. 9: 190192

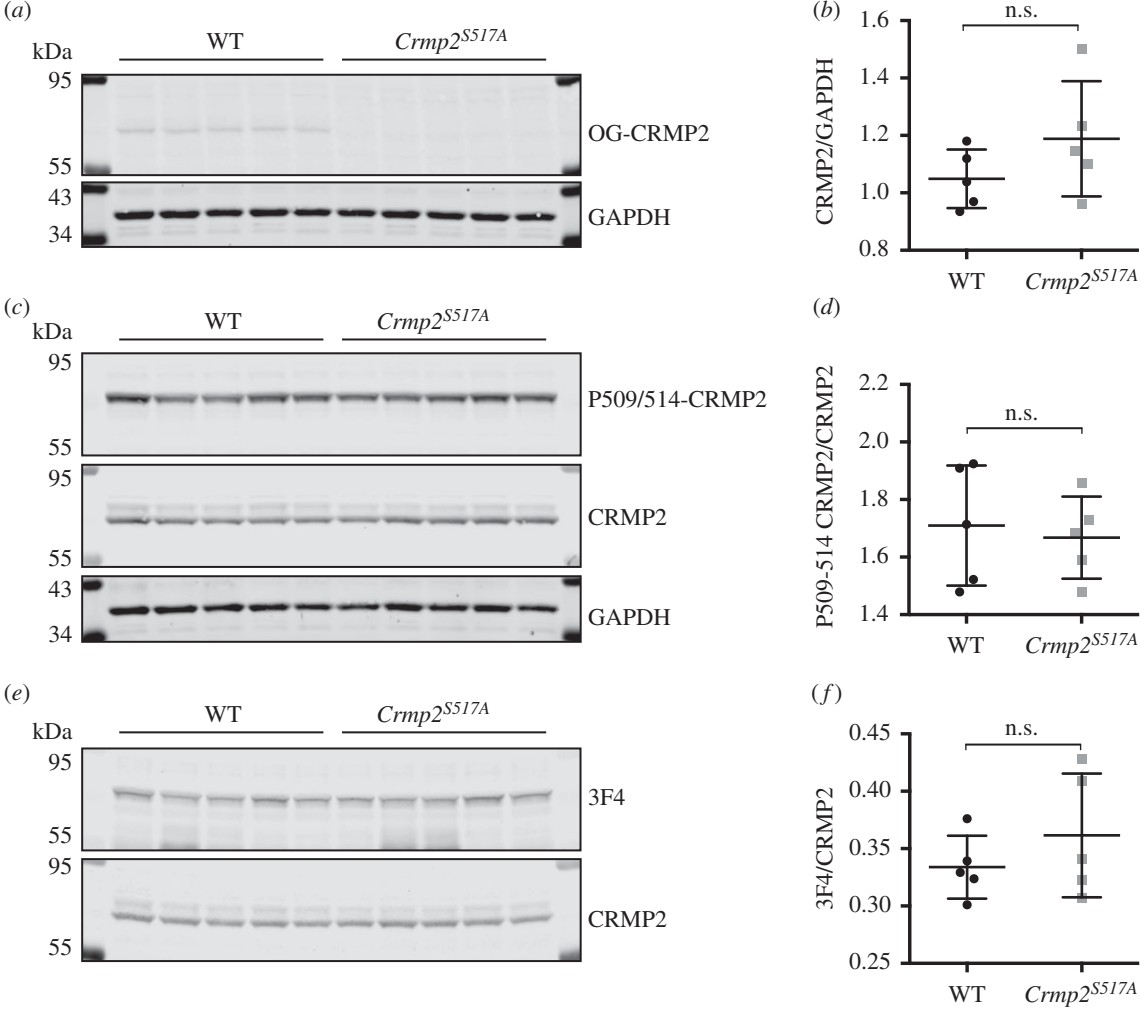

**Figure 2.** CRMP2 protein level and phosphorylation by GSK3β is unchanged in *Crmp2^S517A* mice. (*a*) Mouse brain lysates were probed with site-specific O-GlcNAc-CRMP2 antibody (OG-CRMP2) on western blot. Glyceraldehyde 3-phosphate dehydrogenase (GAPDH) signal was used as loading control. Replicates of five wild-type and *Crmp2^S517A* mice are shown. (*b*) Quantification of CRMP2 signal normalized to GAPDH signal based on blot shown in (*c*). (*c*) Western blot using P509/514-CRMP2 specific antibody and total CRMP2 antibody on mouse brain lysates. (*d*) Quantification of diphospho-CRMP2 signal normalized to total CRMP2 signal. (*e*) Western blot using quadrophospho-CRMP2 (P509/514/518/522) specific antibody, 3F4. (*f*) Quantification of 3F4 signal, normalized to total CRMP2 signal.

## 2.6. *Crmp2^S517A* mice reveal normal gross brain development

Loss of *Crmp2* in mice has been reported to result in abnormal brain development, characterized by enlarged lateral ventricles and aberrant dendrite development [56]. We evaluated gross neuroanatomy in *Crmp2^S517A* and wild-type littermate animals at eight months old from mixed (50%/50%) C57BL/6NTac and C57BL/6 J genetic background (figure 3*c–e*). Conventional histological profiling of *Crmp2^S517A* brains did not reveal any obvious defects in overall neuroanatomy (figure 3*c*). Next, we investigated neuronal and glial cell numbers in the hippocampus using stereological methods. Mutant animals exhibited parameters that were indistinguishable from those of their wild-type control counterparts (electronic supplementary material, figure S6A–C), indicating that CRMP2 O-GlcNAcylation does not affect cell viability in the mammalian nervous system.

CRMP2 is abundant in neurons of the central nervous system and is enriched in axons and synapses [56,74,75]. O-GlcNAcylated proteins, including CRMP2, have been identified on numerous proteins involved in organization and mobilization of synaptic vesicles at nerve termini [60,61,76]. We explored whether O-GlcNAcylation on CRMP2 is

ubiquitous or specific to a cellular compartment or part of the brain. O-GlcNAcylated CRMP2 was detected in fractions of both cytosol and synaptosomes in all sub-dissected regions of the brain, suggesting this modification is ubiquitous (figure 3*f*, *g*). Our data demonstrate that site-specific CRMP2 O-GlcNAcylation at Ser517 is a widespread modification in the nervous system, and is dispensable for gross brain development.

## 2.7. *Crmp2^S517A* mutant mice exhibit selective defects in short-term memory

Missense mutations in *Ogt* lead to intellectual disability, possibly due to changes in the O-GlcNAc proteome [22–26,77]. CRMP2, as one of the most abundant brain O-GlcNAc proteins and a node in axonal guidance signalling, is a candidate effector of this phenotype. To determine whether loss of O-GlcNAc on CRMP2 function affects neuronal function, we conducted comprehensive behavioural profiling of our *Crmp2^S517A* mice. Previous studies have demonstrated that *Crmp2^−/−* knock-out mice exhibit increased locomotor activity, impaired spatial memory formation and abnormal social behaviour [54,56]. Here, we performed novel object recognition (NOR), open-field and spontaneous alternation tests to assess short-term recognition memory, anxiety and spatial

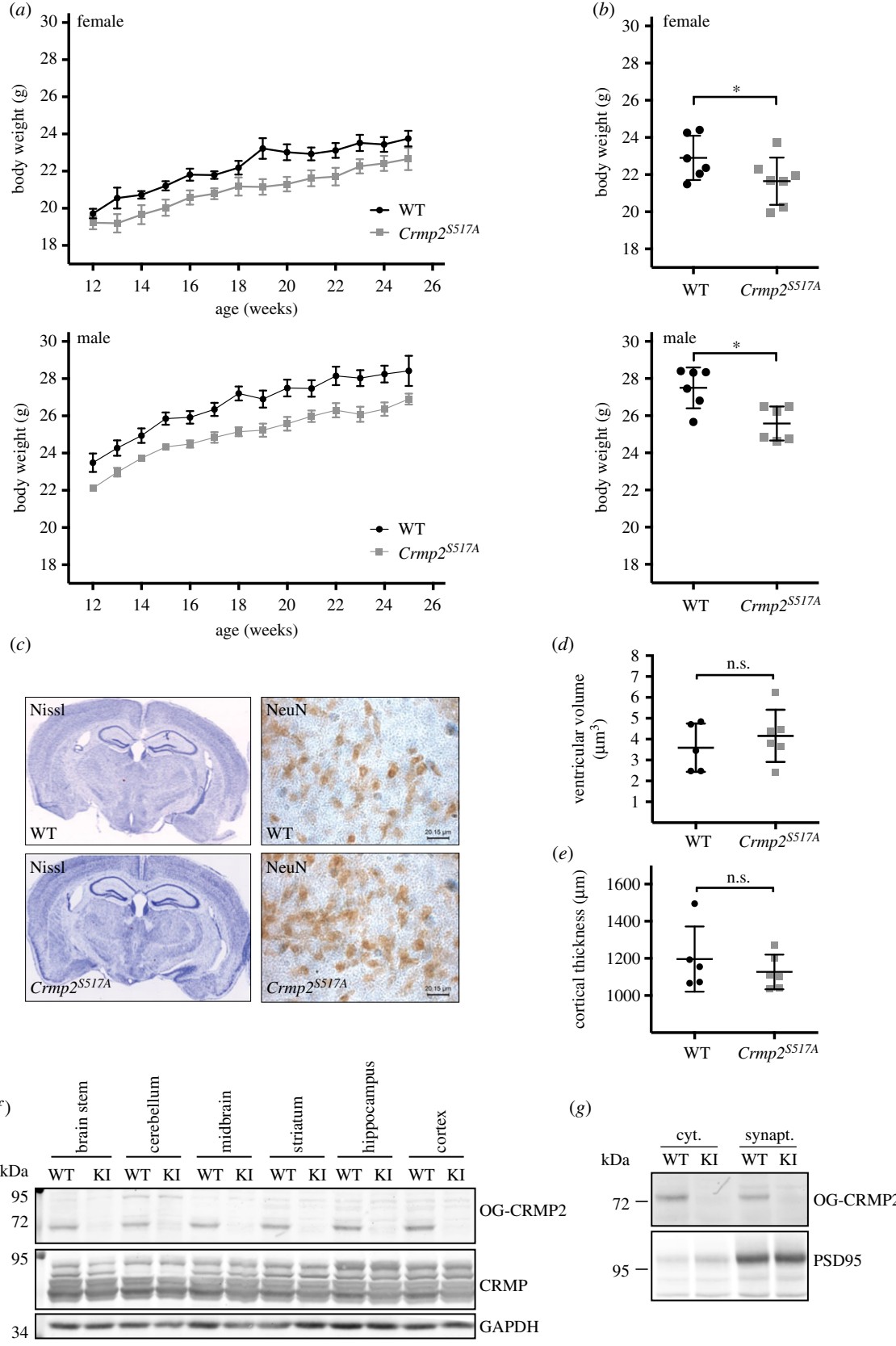

**Figure 3.** Characterization of *Crmp2*^S517A^ mice reveals bodyweight phenotype (*a*,*b*). (*a*) Comparison of the body weights of WT and *Crmp2*^S517A^ mice at 12–25 weeks old. Data are expressed as the mean ± s.e.m. (WT, $n = 8$; *Crmp2*^S517A^, $n = 10$) (male, $n = 9$; female, $n = 9$). Repeated measures ANOVA for the whole dataset, test of between-subject effect, genotype, $F = 11.026$, $p = 0.005$. (*b*) Comparison of the body weights of WT and *Crmp2*^S517A^ mice at 20 weeks old. Data are expressed as the mean ± s.d.; paired, two-tailed *t*-test (female, $p = 0.0199$; WT, $n = 6$; *Crmp2*^S517A^, $n = 7$; male, $p = 0.0125$; WT, $n = 6$; *Crmp2*^S517A^, $n = 6$). Histological analysis of brain samples from WT and *Crmp2*^S517A^ mice (*c–g*). Mean ± s.d.; WT, $n = 5$; *Crmp2*^S517A^, $n = 6$. (*c*) Representative photomicrographs (left panels) show coronal sections stained with cresyl-violet (Nissl staining) in WT and *Crmp2*^S517A^ mice. NeuN immunolabelling (right panels) reveals *Crmp2*^S517A^ has no effect on neuronal viability in the dentate gyrus. (*d*) Ventricular volume (mm³) for left and right lateral ventricles and third ventricle. WT ($3.592 \pm 1.155$) and *Crmp2*^S517A^ ($4.160 \pm 1.252$); n.s. (*e*) Cortical thickness (μm), measurement was taken at four points along the cortex in each animal. WT ($1197 \pm 175.4$) and *Crmp2*^S517A^ ($1127 \pm 93.24$); n.s. (*f*) Western blot shows comparable levels of O-GlcNAc CRMP2 (OG-CRMP2) in several brain regions in WT and *Crmp2*^S517A^ mice relative to total CRMP levels. (*g*) Western blot indicates presence of CRMP2 O-GlcNAcylation in cytoplasmic and synaptic cellular fractions.

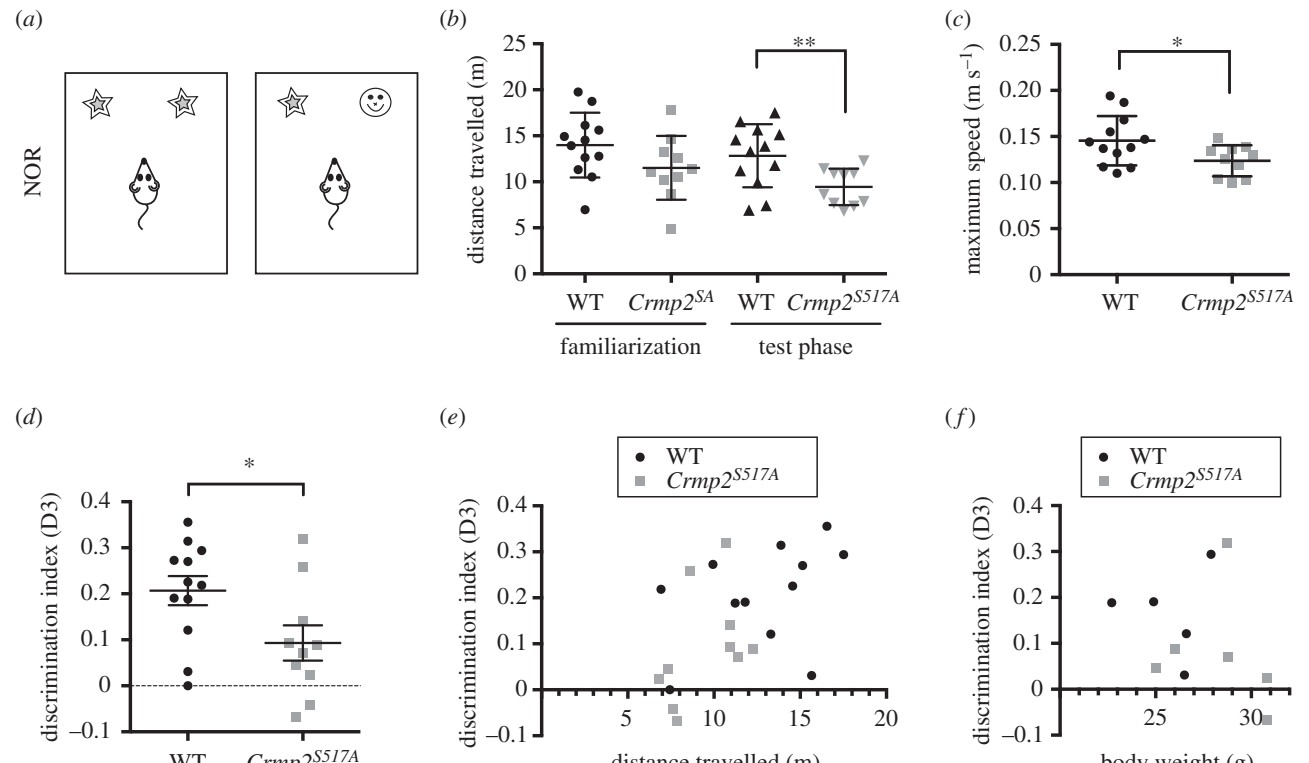

**Figure 4.** $Crmp2^{S517A}$ mice exhibit impaired short-term recognition memory. (d)–(f) Novel object recognition (NOR) test (male, WT, $n = 12$; $Crmp2^{S517A}$, $n = 10$; mean ± s.d.). (a) NOR task. (b) Distance travelled during familiarization and test phase. $Crmp2^{S517A}$ mice (9.4 ± 2.0 m) travelled less distance during the test phase than WT control (12.8 ± 3.4 m) animals; $p = 0.0099$, unpaired $t$-test. (c) Maximum speed reached during the test phase. $Crmp2^{S517A}$ animals (0.124 ± 0.017 m s$^{-1}$) showed slower maximum speed than WT (0.145 ± 0.027 m s$^{-1}$); $p = 0.0327$; unpaired $t$-test. (d) Discrimination index (D3) for WT (0.21 ± 0.11) and $Crmp2^{S517A}$ mice (0.09 ± 0.12); $p = 0.0338$; unpaired $t$-test. (e) Positive correlation between D3 and distance travelled during the test phase; Pearson $r = 0.5344$; $p = 0.0104$; $n = 22$. (f) Correlation analysis between D3 and body weight; Pearson $r = −0.2950$; n.s.; $n = 11$.

working memory, respectively. In addition, locomotor activity was evaluated to identify a possible link to the observed mild weight phenotype. Male $Crmp2^{S517A}$ knock-in mice and wild-type littermate controls ($n = 10–12$ per group) from mixed 12.5% C57BL/6NTac and 87.5% C57BL/6 J genetic background were used at 10–13 weeks old at the start of behavioural testing, reaching 22–25 weeks old at the end of the tests.

First, we performed open-field tests to measure anxiety and simultaneously assess locomotor activity (electronic supplementary material, figure S7A–C). Here, there was no significant difference between $Crmp2^{S517A}$ knock-in mice and wild-type controls in the time spent in the inner/outer zone (electronic supplementary material, figure S7B) and time spent moving in the outer zone (electronic supplementary material, figure S7C), indicating that neither anxiety nor locomotor function of the $Crmp2^{S517A}$ animals are affected.

We next performed an NOR test to assess short-term hippocampal and entorhinal cortex cognitive function (figure 4a–f). This test is based upon the natural instinct of mice to explore novelty, measuring the time the mice explore a novel object versus a familiar object. A calculated discrimination index (D3) of greater than 0.2 indicates that the animal is able to distinguish between the familiar and the novel object. The $Crmp2^{S517A}$ mice, however, completed the NOR task with D3 < 0.2 (figure 4d), significantly lower than wild-type controls (Student's $t$-test, $p < 0.05$), indicating that they were less able to differentiate between new and previously experienced objects. The NOR test also revealed that the $Crmp2^{S517A}$ mice were less mobile during the test phase (Student's $t$-test, $p < 0.01$), travelling shorter distances with lower

maximal speed (figure 4b,c), thus pointing to a possible effect on motivation to explore. Of note, there was a significant positive correlation between the D3 index and distance travelled ($r = 0.53$, $p < 0.05$) (figure 4e). In contrast, there was no correlation between the D3 index and body weight (figure 4f), suggesting that the cognitive phenotype was independent from the weight phenotype.

The observed memory deficit in the NOR task prompted us to investigate memory-related behaviour further. To probe possible deficits in hippocampus-dependent spatial working memory in the $Crmp2^{S517A}$ knock-in mice, we subjected the mice to spontaneous alternation tests in an enclosed plus maze (electronic supplementary material, figure S7D). Individual mice were placed in the centre of the maze and allowed to explore for 15 min. The $Crmp2^{S517A}$ and wild-type mice performed a similar number of total arm entries (electronic supplementary material, figure S7E) and similar percentages of spontaneous alternation (electronic supplementary material, figure S7F), indicating that hippocampus-related spatial memory is not affected but that the observed object recognition memory impairment is likely to be specific to parahippocampal cortical region. Taken together, our study reveals an important association between CRMP2 O-GlcNAcylation status and short-term memory.

# 3. Concluding remarks

Our study sought to decipher the *in vivo* significance of a single site-specific O-GlcNAc modification on CRMP2. Despite the considerable functional redundancy between

members of the CRMP protein family [54,78–80], with multiple post-translational modifications regulating CRMP2 function, we uncovered selective metabolic and neural phenotypes associated with the loss of CRMP2 O-GlcNAcylation.

The phenotypes observed in $Crmp2^{S517A}$ are more modest and selective than those of both the $Crmp2$ knock-out and the $Crmp2^{S522A}$ knock-in models that are deficient in Cdk5/GSK3β-mediated phosphorylation. While $Crmp2^{-/-}$ knock-out mice have increased lateral ventricles [56], gross brain development appeared normal in both $Crmp2^{S522A}$ [81] and our $Crmp2^{S517A}$ mutant mice reported here. In terms of motor function and cognition, there was little overlap between those observed in $Crmp2^{S517A}$, $Crmp2^{-/-}$ and $Crmp2^{S522A}$ models. $Crmp2^{-/-}$ mice showed increased locomotion [55,56], while locomotion was not affected in $Crmp2^{S522A}$ mice [82]. Interestingly, our $Crmp2^{S517A}$ mutant animals exhibited signs of altered locomotion during the NOR task, exhibiting less movement at lower speed specifically in the test phase. Despite this selective phenotype, such defects were not present in the plus maze and open-field tests.

Intriguingly, learning and short-term memory were selectively impaired in $Crmp2^{S517A}$ mice, as revealed in the NOR task, which requires orchestrated input from several brain regions including insular cortex, perirhinal cortex, ventromedial prefrontal cortex and hippocampus [83]. In contrast to our knock-in observations, cognitive impairments in $Crmp2^{-/-}$ mice are primarily hippocampus-centric, where CRMP2 is highly expressed [84]. Spatial and contextual learning abilities were also reduced in $Crmp2^{-/-}$ mice, yet such a direct comparison with these previous studies is challenging, owing to background and experimenter variability.

We detected a reduction in body weight in our first cohort of $Crmp2^{S517A}$ mice, but not in the second cohort with a higher (87.5%) contribution of the C57BL/6 J background. The C57BL/6 J variant mice contain a deletion within the nicotinamide nucleotide transhydrogenase ($Nnt$), a nuclear gene that encodes a mitochondrial inner membrane protein that maintains intramitochondrial redox homeostasis. Loss of $Nnt$ is associated with numerous phenotypic consequences, including altered glucose homeostasis [85] and weight gain in male mice [86]. Moreover, loss of $Nnt$ is known to modify behavioural and neuroendocrine regulation in mice [87] and also influences the severity of phenotypes arising from pathological mutations [88]. Similarly to our observation, the brain-specific $Crmp2^{-/-}$ knock out mice showed a reduction in weight [55,56] whereas the total $Crmp2^{-/-}$ deficient mice did not [55], suggesting that this phenotype might manifest only in certain genetic backgrounds.

The $Crmp2^{S517A}$ mice displayed behavioural phenotypes distinct from those observed in $Crmp2^{-/-}$ or $Crmp2^{S522A}$ mice, suggesting the Ser517Ala mutation neither blocks CRMP2 function severely nor affects Ser522 phosphorylation. Owing to the close proximity of Ser517 to Cdk5/GSK3β phosphosites and previous peptide-based studies predicting phosphorylation/O-GlcNAcylation interplay, we hypothesized that the Ser517Ala mutation would alter CRMP2 phosphorylation, with potential effects on microtubule interactions and downstream neuronal phenotypes. However, neither behavioural nor biochemical characterization has provided evidence that the observed phenotypes are underpinned by altered phosphorylation status *in vivo*. It is conceivable that compensatory alterations in expression and activity of other CRMP family members [55,82] could mask any direct effects of the Ser517Ala mutation.

Ser517 resides in the structurally flexible C-terminal tail of CRMP2 required for stabilizing homo/hetero CRMP tetramers [89,90]. A crystal structure of the CRMP2 tetramer with a partially truncated C-terminal tail (CRMP2 1-525) revealed extensive intermolecular interactions involving the C-terminus [89]. It has also been shown that the C-terminal tail mediates binding to the microtubule network [89]. Therefore, a possible molecular mechanism underpinning the behavioural phenotypes is that O-GlcNAcylation on the CRMP2 C-terminal tail could affect tetramer formation, stability and microtubule-lattice binding, which will require further investigation. Taken together, our data support a robust regulatory network for CRMP2 function and a fine-tuning role for O-GlcNAcylation on Ser517 that plays an important role in memory formation and neuronal plasticity in mammals.

Altered protein O-GlcNAcylation has been proposed as a possible contributing factor to the development of AD. Hyperphosphorylation of CRMP2 has been observed in neurofibrillary tangles in AD brains [41]. Studies on mouse models of AD have demonstrated that CRMP2 hyperphosphorylation is an early event during the development of AD and occurs prior to plaque and tangle formation [41,42]. This inspired us to study how O-GlcNAcylation potentially regulates phosphorylation on CRMP2 and whether the stoichiometry of O-GlcNAcylated versus unmodified CRMP2 is reduced in AD. We did not find a significant increase of O-GlcNAcylation on CRMP2 in AD compared with age-matched individuals. In addition, no large global changes in CRMP2 phosphorylation were detected *in vivo* owing to the absence of O-GlcNAcylation. However, detection of such changes is challenging because of the potentially low stoichiometry of CRMP2 O-GlcNAcylation in tissue.

For over 30 years, functional dissection of protein O-GlcNAcylation has been mostly limited to genetic and pharmacological manipulation of OGT and OGA, key enzymes responsible for O-GlcNAc cycling. These studies have highlighted the role of O-GlcNAc in several physiological contexts. Importantly, research on cell culture models has also uncovered the function of O-GlcNAc on a few defined target proteins in specific cellular processes. Despite this large body of work, research on the significance of O-GlcNAcylation on discrete sites of specific proteins *in vivo* is scarce. Such data are ultimately required for a complete mechanistic understanding of the phenotypes observed in $Oga^{-/-}$ or $Ogt^{-/-}$ models and human disease states associated with disrupted O-GlcNAc homeostasis. Our findings constitute an important advance towards a protein-specific understanding of O-GlcNAcylation in the mammalian nervous system. Future studies will be essential to decipher the complex interplay between O-GlcNAcylation and other post-translational modifications.

# 4. Material and methods

## 4.1. Antibodies

Rabbit anti-gSer517 OG-CRMP2 (1 : 1000) raised for this study by the van Aalten laboratory

Rabbit anti-CRMP2 (1 : 2000) Millipore

royalsocietypublishing.org/journal/rsob   Open Biol. 9: 190192

royalsocietypublishing.org/journal/rsob Open Biol. 9: 190192

Rabbit anti-CRMP2 (1 : 2500) Cell Signalling 9393

Rabbit anti-pan-TUC (CRMP) (1 : 10 000) Millipore ABN108

Sheep anti-P-T509/514 CRMP2 (1 : 2000) with dephospho-peptide [58]

Mouse anti-O-GlcNAc /RL2 (1 : 2000) Abcam ab2739

Mouse anti-FLAG (M2) (1 : 5000) Sigma

Mouse anti-PSD95 (SAP90, DLG4) (108E10) SYSY

Rabbit anti-GAPDH (FL-335) (1 : 1000) Santa Cruz sc-25778

Goat anti-GATA-2 (0.5 µg ml$^{-1}$) R&D Systems AF2046

Mouse 3F4 (1 µg ml$^{-1}$) Takara 29060

Mouse NeuN (A60) (1 : 2000) Millipore MAB377

Licor 680, Licor 800

## 4.2. Purification of CRMP from brain tissue

CRMPs were purified from sheep brain by employing three rounds of ion exchange chromatography protocol as previously described [91]. Briefly, sheep brain tissue (100 g) was homogenized in three volumes of DEAE buffer (25 mM sodium phosphate (pH 7.8) containing 2 mM dithiothreitol (DTT) and 1 mM phenylmethylsulfonyl fluoride) using an Emulsiflex continuous flow cell disruptor (Constant). The tissue lysate was centrifuged at 100 000$g$ for 45 min at 4°C. The supernatants from two centrifugations were combined and applied to a 10 ml DEAE-Sephadex column (Sigma) equilibrated with DEAE buffer. After several washes with DEAE buffer, bound proteins were eluted with a 100, 200 and 300 mM NaCl step gradient in DEAE buffer. Peak CRMP-containing fractions in the 200 mM NaCl elution were combined, diluted 10-fold with S buffer (25 mM sodium phosphate (pH 6.0) containing 2 mM DTT) and applied to a 5 ml SP-Sepharose column equilibrated with S buffer. After washing, proteins were eluted with 150 mM NaCl in S buffer. Peak CRMP-containing fractions were pooled and diluted fivefold with HA buffer (10 mM potassium phosphate (pH 7.0) containing 2 mM DTT). The diluted sample was applied to a 3 ml hydroxyapatite column (Bio-gel HTP; Bio-Rad) and eluted with a linear gradient of 10–100 mM potassium phosphate (pH 7.0) containing 2 mM DTT, and peak CRMP-containing fractions were pooled. All steps were performed on ice or at 4°C. CRMP protein purification was monitored by anti-CRMP immunoblotting.

## 4.3. Enrichment of O-GlcNAc-CRMPs

GST-$Cp$OGA$^{D298N}$ was expressed in BL21(DE3)pLysS *Escherichia coli* and cells were suspended in 25 mM Tris/250 mM NaCl, pH 7.5 buffer containing 0.1 mg ml$^{-1}$ DNAse, 1 mg ml$^{-1}$ lysozyme and protease inhibitors 1 mM benzamidine, 0.2 mM PMSF and 5 µM leupeptin, and lysed by homogenization with an EmulsiFlex-C3 (Avestin). After centrifugation at 40 000$g$ for 20 min, the supernatant containing GST-$Cp$OGA$^{D298N}$ was bound to Glutathione Sepharose 4B beads for 45 min at 4°C and washed with 25 mM Tris/ 250 mM NaCl, pH 7.5. Purified sheep CRMP2 (6 mg) was exposed to GST-$Cp$OGA$^{D298N}$ bound to beads for 90 min at 4°C in 1 ml of phosphate-buffered saline (PBS), 0.01% CHAPS. O-GlcNAcylated protein was eluted with 0.2 ml 0.75 M GlcNAc for 2 × 5 min. A total output of approximately 1 µg glyco-enriched CRMP2 was obtained.

## 4.4. O-GlcNAc site mapping of CRMP2

Sample preparation for mass spectrometry: 500 ng of glyco-enriched CRMP2 was subjected to 10% sodium dodecylsulfate–polyacrylamide gel electrophoresis (SDS-PAGE) in a concentrated band of 0.5 cm, followed by manual enzymatic digestion as previously reported [92] with minor modifications. Briefly, excised bands were rinsed three times with AmBic buffer that consisted of 50 mM ammonium bicarbonate in 50% methanol (high-performance liquid chromatography (HPLC) grade; Merck) following a reduction step with 10 mM DTT (Sigma-Aldrich). Subsequently, the gel pieces were rinsed twice with AmBic buffer and dried in a SpeedVac before alkylation with 55 mM iodoacetamide (Sigma-Aldrich) in 50 mM ammonium bicarbonate. The gel pieces were rinsed with AmBic buffer before being dehydrated by addition of acetonitrile (HPLC grade; Merck) and drying in a SpeedVac. Trypsin (Promega, Madison, WI) was added to the dry gel pieces at a final concentration of 20 ng µl$^{-1}$ in 20 mM ammonium bicarbonate, and then incubating them at 37°C for 16 h. Peptides were extracted three times by 20 min incubation in 40 µl of 60% acetonitrile in 0.5% HCOOH. The resulting peptide extracts were pooled, concentrated in a SpeedVac and stored at −20°C until analysis by mass spectrometry.

Identification of the O-GlcNAcylated proteins and mapping of O-GlcNAc sites was performed by electrospray ion trap electron transfer dissociation (ESI-IT-ETD) mass spectrometry coupled to a nano-LC system (Ultimate 3000 RSLC; Dionex, Netherlands). Dried peptides were resuspended in 30 µl of 0.5% HCOOH and 10 µl was injected for mass spectrometric analysis. Tryptic peptides were concentrated on a trap column (2 cm × 100 µm, Dionex) at 10 µl min$^{-1}$ and separated on a 15 cm × 75 µm Pepmap C18 reversed-phase column (Thermo Fischer Scientific). Peptides were eluted by a linear 60 min gradient of 95% A/5% B to 90% B (A: H$_2$O, 0.1% HCOOH; B: 80% acetonitrile, 0.08% HCOOH) at 300 nl min$^{-1}$ into an LTQ Velos ETD (Thermo Fisher Scientific). Mass spectrometry spectra were acquired in positive mode: firstly mass spectrometer full scans were acquired followed by MS/MS in ETD mode. Up to 10 of the most intense precursors were selected for ETD fragmentation with an activation time of 300 ms and non-dynamic exclusion. Proteome Discoverer v 1.4.0.288 software (Thermo Scientific) was used to process raw LC-MS/MS data, applying the Mascot (version 2.4; Matrix Science, Boston, MA, USA) search engine algorithm against the SwissProt database with the following Mascot parameters: 2+, 3+, 4+ and 5+ ions; precursor mass tolerance 10 ppm; Da; fragment tolerance 0.6 Da and up to two missed cleavages. The variable modifications included were: oxidation (M) (15.99 Da), dioxidation (M) (31.98 Da) and HexNAc (ST) (+203.0794 Da). All MS/MS data and database results were manually inspected to verify accurate assignment of fragment ions using the above software. Peptides with *E*-value < 0.1 are considered as a precise O-GlcNAc site assignment.

## 4.5. Phosphorylation analysis of CRMP from mouse brain tissue samples

CRMPs were purified from single mouse brain samples one by one by employing two rounds of ion exchange chromatography. We followed the same protocol as for CRMP purification from sheep brain described earlier without carrying out the third chromatography step on the hydroxyapatite

column. For phosphopeptide analysis, purified CRMP2 was first reduced and alkylated by addition of 10 mM TCEP and 40 mM chloroacetamide. CRMP2 was then digested by addition of LysC (1 : 100 ratio LysC : CRMP2) followed by trypsin (1 : 25 ratio trypsin : CRMP2) at 37°C overnight. The resulting peptides were desalted using Sep-Pak cartridges (Waters) and phosphopeptides enriched using Fe-IMAC according to a previously described procedure [93]. LC-MS/MS analysis was performed on an Agilent 1290 UPLC system (Agilent) coupled to an Orbitrap Fusion Tribrid mass spectrometer (Thermo Fisher Scientific). The phosphopeptides were trapped and eluted using a gradient of 8–44% B over 70 min where buffer A was $H_2O$ containing 0.1% formic acid and buffer B was 80% acetonitrile with 0.1% formic acid. EThcD was used to fragment the peptides with 40% supplemental activation set and an activation time of 50 ms. Raw data were analysed using Byonic software (Protein Metrics Inc.) using the following parameters: precursor ion tolerance 10 ppm, product ion mass tolerance 20 ppm. Cys carbaminomethylation was set as a fixed modification and Met oxidation and STY phosphorylation as variable modifications. The number of mis-cleavages was set to 5. The phosphosites identified were manually confirmed to ensure 100% confidence for phosphosite localization.

## 4.6. Generation of O-GlcNAc-specific antibody against Ser517 on CRMP2

A polyclonal O-GlcNAc-specific antibody against Ser517 on CRMP2 was generated using a previously described approach [94] with some modifications. The O-GlcNAc peptide CKTVTPA[O-GlcNAc]SSAKTSPA and the matching unmodified peptide corresponding to residues 511–524 of human CRMP2 were synthesized on a Liberty microwave-assisted peptide synthesizer (CEM) using MBHA Rink-amide low-load resin (Novabiochem) with standard protocols of Fmoc SPS chemistry. 4Ac-GlcNAcSerFmoc was synthesized in-house following a published procedure [95]. The peptides were purified using HPLC and conjugated to keyhole limpet haemocyanin (KLH) prior to injection into rabbits. Antibodies from the serum were purified following a two-step affinity purification protocol, first over a non-GlcNAcylated peptide column, then subsequently over a GlcNAcylated peptide column (Dundee Cell Product). The non-GlcNAc-CRMP2 antibody fraction was retained on the first column and the GlcNAc-CRMP2-specific antibodies were collected in the flow through. This fraction was then loaded onto a GlcNAc-CRMP2 peptide column to purify the GlcNAc-CRMP2-specific antibody from other immunoglobulins. We probed antibody specificity using recombinant CRMP2, where it recognized the in vitro O-GlcNAcylated CRMP2 but not the non-glycosylated or the Ser517Ala point mutant CRMP2 (electronic supplementary material, figure S2A). Gratifyingly, this antibody was also able to recognize O-GlcNAc CRMP2 as a single band on western blot in mouse brain lysate (electronic supplementary material, figure S2B). We further explored antibody specificity on mouse brain lysate treated with CpOGA, a bacterial O-GlcNAcase from Clostridium perfringens, possessing a high-level hydrolytic activity on eukaryotic O-GlcNAc proteins. The antibody did not react with naked CRMP2 as removal of all O-GlcNAc modification from proteins abolished its detection.

## 4.7. DNA cloning

C-terminally truncated CRMP2 1-536 was inserted into pGEX6P1 vector, containing an ampicillin resistance cassette, for recombinant protein production in E. coli and into pRK-Flag plasmid for transfection of mammalian cells. The Ser517Ala mutation in CRMP2 was introduced following the QuickChange method (Stratagene) using KOD Hot Start Polymerase (Novagen) and verified by sequencing. Constructs for CpOGA and GST-CpOGA$^{D298N}$ (amino acids 31–618) expression were described before [62,96].

## 4.8. Protein expression and purification

GST-tagged CRMP2 (amino acids 1–536) was expressed in BL21 E.coli and purified on a Glutathione Sepharose 4B column and cleaved using PreScission Protease. Recombinant OGT was produced as previously described [65]. Expression of CpOGA and GST-CpOGA$^{D298N}$ was reported previously [62,96].

## 4.9. In vitro O-GlcNAcylation and CpOGA reaction

Purified recombinant CRMP2 was O-GlcNAcylated in vitro in a reaction mixture containing 250 nM recombinant human OGT, 1 µM CRMP2, 100 µM UDP-N-acetylglucosamine incubated for 60 min at 37°C. Aliquots of 100 µg of mouse brain lysates were incubated with 15 µg of CpOGA for 60 min at 30°C in order to remove O-GlcNAc modification and to elucidate the specificity of the gSer517-CRMP2 antibody.

## 4.10. Cell culture

The Neuro-2a mouse neuroblastoma cell line was maintained in Dulbecco's modified minimal essential medium (DMEM) (Life Technologies, Inc., Burlington, ON) supplemented with 10% (v/v) fetal calf serum (Life Technologies, Inc.), L-glutamine (2 mM), penicillin (100 units ml$^{-1}$) and streptomycin (100 µg ml$^{-1}$; Life Technologies, Inc.) at 37 °C, under 5% $CO_2$. Cells were transfected using polyethylenimine (PEI) using a 3 : 1 ratio of PEI to DNA (w/w) diluted in serum-free DMEM. After 48 h of transfection, the cell culture medium was changed and cells were treated with 25 µg ml$^{-1}$ cycloheximide (CHX) alone for 6 h then supplemented with 1 µM GlcNAcstatin G (GlycoBioChem) for an additional 16 h.

## 4.11. Immunoprecipitation with anti-Flag antibody

Cell lysates obtained from Neuro2a cells expressing Flag-CRMP2 were incubated with anti-Flag M2 agarose beads (Sigma, A2220) overnight in the cold room. Beads were washed with cold PBS three times and Flag-CRMP2 was eluted with SDS sample buffer; 1% 2-mercaptoethanol was added prior to running the samples on SDS-PAGE gels.

## 4.12. Western blotting

Mouse brain tissue was rapidly dissected, rinsed in cold PBS, snap frozen in liquid nitrogen and stored at −80°C. For immunoblotting cells/tissue were lysed in 50 mM Tris-HCl pH 7.4, 0.1 mM EGTA, 1 mM EDTA, 1% Triton-X100, 1 mM sodium orthovanadate, 50 mM sodium fluoride, 5 mM sodium pyrophosphate, 0.27 M sucrose, 0.1% 2-mercaptoethanol supplemented with protease inhibitors (1 mM benzamidine,

royalsocietypublishing.org/journal/rsob Open Biol. 9: 190192

0.2 mM PMSF and 5 µM leupeptin) and 10 µm GlcNAcstatin G. For *Cp*OGA treatment GlcNAcstatin G was omitted from the lysis buffer. Cell lysate was centrifuged at 14 000 rpm for 20 min at 4°C and the protein concentration was determined with the Pierce 660 nm protein assay. A total of 20–30 µg of protein was denatured in SDS loading buffer containing 1% 2-mercaptoethanol or 200 mM DTT. Proteins were separated on precast 8% NuPAGE Bis-Tris Acrylamide gels (Invitrogen) by SDS-PAGE and transferred to nitrocellulose membrane. Membranes were incubated with primary antibodies in blocking buffer, 5% bovine serum albumin in TBST (Tris-buffered saline with 0.1% Tween-20) overnight at 4°C and next with IR680/800-labelled secondary antibodies at room temperature for 1 h. Blots were imaged using the Li-Cor Odyssey infrared imaging system (Li-Cor) and Image Studio Lite software.

## 4.13. Sample preparation from human post-mortem tissue

Human post-mortem tissue samples (control $n = 10$ hippocampus, age range 17–89, AD $n = 5$ hippocampus, age range 75–80) were obtained from the MRC London Neurodegenerative Disease Brain Bank and the MRC Sudden Death Brain and Tissue Bank, University of Edinburgh, UK. Frozen tissue was hand-homogenized in a glass Dounce homogenizer in ice-cold standard tissue lysis buffer (50 mM Tris-HCl pH 7.4, 1% Triton X-100, 0.1 M EGTA, 1 mM EDTA, 1 mM $Na_3VO_4$, 50 mM NaF, 5 mM Na PyroP, 0.27 M sucrose, 0.1% β-mercaptoethanol containing Complete Protease Inhibitor Cocktail), followed by centrifugation at $20\,000g$ (15 min, 4°C) and the supernatant collected.

## 4.14. GlcNAcstatin G continuous infusion

Alzet osmotic pumps (model: 2001D) loaded with GlcNAcstatin G (200 µl formulated in 50% DMSO, 25% PEG400, 25% sterile water; 10 mg ml$^{-1}$; $n = 4$ mice) or vehicle alone (200 µl formulated in 50% DMSO, 25% PEG400; 25% sterile water; $n = 3$ mice) were implanted subcutaneously between the shoulder blades of anaesthetized female NMRI mice. A cannula from the pump was inserted into the jugular vein. Mice were housed in standard holding cages with water and food available ad libitum and 12 h/12 h light/dark cycle throughout the study. At 24 h post implant a terminal blood sample was taken from each mouse and brain was harvested for biochemical analysis. One mouse from the GlcNacstatin G-infused group was used for pharmacokinetics analysis only. Spot blood samples (10 µl) were taken from this mouse at 15, 30 min, 1, 2, 4, 6 and 21 h, followed by a terminal blood sample at 24 h and harvesting of brain for bioanalysis (electronic supplementary material, table S1).

## 4.15. Generation of *Crmp2*$^{S517A}$ mice

*Crmp2*$^{S517A}$ mice were generated by TaconicArtemis GmbH (electronic supplementary material, figure S4). Briefly, the CRMP2 coding sequence spanning exons 12–14 was targeted by homologous recombination in C57BL/6NTac (Art B6 3.6) mouse embryonic stem cells. The construct coding for the Ser517Ala also contained an FRT-flanked puromycin resistance cassette in the intronic region between intron 13 and 14, allowing for selection for recombination events (electronic

supplementary material, figure S4). A total of 10–15 cells from selected ES colonies bearing the Ser517Ala mutation were injected into 3.5-day-old blastocyst embryos of BALB/c mice. After recovery, embryos were transferred to pseudo-pregnant NMRI female mice. Chimeric progenies were crossed with Flp deleter C57BL/6-Tg(CAG-Flpe)2 Arte mice and heterozygous pups carrying the Ser517Ala mutation but lacking the puromycin resistance cassette were selected. These founder heterozygous mice were crossed with C57BL/6 J wild-type animals. The line was initially bred by inter-crossing heterozygous animals, then maintained by backcrossing to the C57BL/6 J genetic background.

Genotyping was performed by diagnostic polymerase chain reaction using KOD Hot Start DNA polymerase (EMD Millipore) on genomic DNA isolated from tissue biopsy with forward 5′-TTGTTAAAGAATGTGGAACTGGG-3′ and reverse 5′-AAGTGTTCCTCATTCCTCATGG-3′ primers that amplified a 385 bp fragment of the knock-in or a 310 bp fragment of the wild-type allele. All animal studies and breeding were performed in accordance with the Animal Scientific Procedures Act of 1986.

## 4.16. Histology and immunohistochemistry

Eleven animals, five wild-type and six homozygous *Crmp2*$^{S517A}$ mice, were sacrificed for assessing gross brain development. Mice under terminal anaesthesia (Euthatal 9 mg kg$^{-1}$) were subjected to cardiac perfusion firstly with PBS and then with 4% paraformaldehyde. Brains were quickly dissected and post-fixed by immersion in 4% paraformaldehyde in 200 mM phosphate buffer pH for overnight fixation. Following a brief wash in PBS, they were placed in 25% sucrose. We ensured an approximately 20-fold volumetric excess of solution to tissue at all stages.

In order to eliminate bias, samples were blinded and then processed and scored by an independent researcher. A 1 : 12 series of sections was used for Nissl staining using cresyl violet. A further 1 : 12 series was processed for immunohistochemistry with mouse NeuN (A60, Millipore, MAB377, 1 : 2000) primary antibody specific to neurons and visualized with horseradish peroxidase-linked secondary antibody (1 : 200).

## 4.17. Subcellular fractionation

The synaptoneurosome fraction was purified as previously described [97]. Briefly, mouse brain tissue samples were collected and snap frozen in liquid nitrogen. The tissue was later homogenized in a pre-chilled Dounce homogenizer with 2 ml ice-cold buffer A (25 mM HEPES (pH 7.5), 120 mM NaCl, 5 mM KCl, 1 mM $MgCl_2$ and 2 mM $CaCl_2$) supplemented with 2 mM DTT, protease inhibitors (1 mM benzamidine, 0.2 mM PMSF and 5 µM leupeptin) and phosphatase inhibitors (1 mM $Na_3VO_4$, 50 mM NaF and 5 mM Na PyroP). The lysate was cleared by passing it through two layers of 80 µm nylon filters (Millipore, NY8002500) to remove tissue debris. Total extract sample was prepared at this stage. The synaptoneurosomes fraction was purified by passing the homogenate through a 5 µm Supor membrane filter (Pall Corp., Port Washington, NY) and then spinning down at $1000g$ for 5 min. This yielded the supernatant used for obtaining the cytosol fraction and a pellet containing synaptoneurosomes. The supernatant was subjected to a 1 h

centrifugation at 100 000$g$ to clear the cytosol fraction. The synaptoneurosome pellet was further washed once with buffer A and centrifuged. The pellet was resuspended in 0.5 ml buffer B (50 mM Tris pH 7.5, 1.5% SDS, 2 mM DTT), and boiled for 5 min.

## 4.18. Body weight measurement

Heterozygous pairs of $CRMP2^{S517A/+}$ animals on a 50%/50% C57BL 6 J/6NTac mixed genetic background were used to obtain experimental animals for body weight measurements. Wild-type and homozygous $Crmp2^{S517A}$ mice were housed together in the same cage in groups and fed with standard chow. Body weights of approximately 13 male and approximately 13 female mice were followed weekly between three and six months of age. Data collection was performed by an independent observer in a blinded manner.

One out of seven wild-type male mice had an unusually low body weight compared with the wild-type dataset (20.39 g at 12 weeks, 22.66 g at 16 weeks, 23.38 at 20 weeks and 24.14 g at 24 weeks of age), otherwise no illness was reported. We have excluded this mouse from our data analysis.

## 4.19. Behavioural testing

Prior to any behavioural testing, all animals experienced daily handling by the experimenter for two weeks. This process ensured that the animals were accustomed to being picked up and handled by the experimenter and vice versa. All handling took place in the experimental room at the same time each day. Male $Crmp2^{S517A}$ knock-in and wild-type littermate control ($n = 8$–12 per group) animals from 12.5% C57BL/6NTac and 87.5% C57BL/6 J genetic background participated in the study. All animals were housed in groups. Animals were 10–13 weeks old at the start of behavioural testing, reaching 22–25 weeks old at the end. The experiment was performed blind, with the experimenters unaware of genotype while performing the test and analysing the data.

## 4.20. Novel object recognition test

Cognition was assessed by the NOR test. During the familiarization phase, the animal is exposed to two identical objects for 10 min. Following an inter-phase interval for 10 min, the animal is exposed to a single copy of the familiar object and a novel object (test phase). Discrimination index (D3) was calculated [98]:

$$D3 = \frac{\text{(time spent exploring novel object } - \text{ time spent exploring)}}{\text{total time spent exploring}}.$$

## 4.21. Open-field test

The activity of mice in an open-field maze was recorded. Time spent in the centre and periphery and time moving in the periphery were analysed to investigate parameters of locomotion and anxiety. The test started by placing the mice in the middle of the open-field cage and each animal was allowed to explore for 15 min.

## 4.22. Spontaneous alternation test

The spontaneous alternation test performed in an enclosed plus maze was used to assess spatial working memory. Individual mice were placed in the centre of the maze and allowed to explore for 15 min. Each arm entry was recorded to calculate the percentage of spontaneous alternation. Mice that were able to remember which arms they had entered most recently would choose a different one to explore,

$$\% \text{ of alternations} = \frac{\text{total no. alternations}}{\text{(total entries } - 5)}.$$

## 4.23. Statistical analysis

Statistical analyses were performed with Prism 6 and SPSS software packages. For pairwise comparisons of wild-type and $Crmp2^{S517A}$ data the Student's $t$-test was used. Unpaired $t$-tests were calculated for independent samples; paired $t$-tests were used when littermate correlation was apparent. For body weight data, ANOVA with repeated measures was performed. Test of between-subject effects indicated the significant effect of genotype on body weight.

Ethics. All animal studies and breeding in Dundee were approved by the University of Dundee ethics review committee, and further subjected to approved study plans by the Named Veterinary Surgeon and Compliance Officer (Dr Ngaire Dennison) and performed under a UK Home Office project licence in accordance with the Animal Scientific Procedures Act (ASPA, 1986).

Data accessibility. This article has no additional data.

Authors' contributions. V.M. and D.M.F.v.A. conceived the study; V.M., R.H., R.W., T.G.M. and M.S. planned and performed experiments; J.A. and A.C.L. performed and analysed mass spectrometry; A.D.M. performed and analysed behavioural studies; K.D.R. oversaw the DMPK studies, V.M. analysed and interpreted the data and V.M., R.W. and D.M.F.v.A. wrote the manuscript with input from all authors.

Competing interests. The University of Dundee holds a patent for the GlcNAcstatin G inhibitor.

Funding. This work was funded by a Wellcome Trust Senior Research Fellowship (WT087590MA) to D.M.F.v.A., an ARUK Pilot Project grant to R.W., and support from Tenovus Scotland to V.M. The phosphoproteomics mass spectrometry work was supported by the Horizon 2020 program INFRAIA project Epic-XS (project 823839) to A.J.R.H.

Acknowledgements. We thank Vladimir Borodkin and Andrew Ferenbach for their invaluable contribution with peptide synthesis and molecular biology, respectively. We thank and acknowledge Frederick Simeons and Laste Stojanovski for carrying out the osmotic pump studies. We thank Arron Dow for performing the animal behavioural studies. We also thank our departmental support teams for their assistance (Medical Research Council Genotyping and Tissue Culture teams) and the School of Life Sciences Biological Services (all resource units) for the essential management, maintenance and husbandry of mice.

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
