## [Reviewer comments · Open Biology]

Review History

RSOB-19-0192.R0 (Original submission)

Review form: Reviewer 1

Recommendation

Accept with minor revision (please list in comments)

Do you have any ethical concerns with this paper?

No

Comments to the Author

1. By other reports, it is known that O-GlcNAcylation is decreased in AD patients, and as such increased O-GlcNAc is hypothesised to confer protective effects in the context of AD (Tau for example). Can you elaborate on the observation that O-GlcNAcylation of CRMP-2 is not statistically different from normal to AD? Would this data look different if this was performed in a CRMP3 (also hippocampal in mature neurons) knockdown background in a cell line (WT versus AD) that may remove any compensation to tease out the role of the S517 of CRMP2? If

most CRMPs are not present in the adult brain, how exactly is this compensation mechanism in play in the 3-6 month old adult mice?

2. Are any of the other CRMPs O-GlcNAcylated?
3. Is there any available information on known SNP variants of CRMP2 that are correlated in any neurodegenerative diseases? Are any of the S/T sites discussed in this report known to segregate with disease?
4. O-GlcNAcylation plays a role in protein-protein interactions. Knowing that CRMP2 and Tau are both involved in neurofibrillary tangles, and associated with each other, do any of the other known O-GlcNAc modified proteins in microtubules interact with CRMP2? Is this interaction lost with the 517A mutant thereby contributing to or causing the effects seen with the phenotypes seen?
5. Considering that the authors are drawing parallel (as inferred from ancillary data) to healthy versus neurodegeneration (in this case AD) of aging human brains, the more appropriate age group for the mouse study should be 14+ months. Is there any data available for mice older than 6 months?
6. Several of the western blot images do not show molecular weight markers in the supplementary figures, Fig. 3G and 1E.
7. Do the authors have an explanation that when tested can elaborate on the weight loss phenotype knowing the relation of O-GlcNAc and OGT to nutrient sensing, glucose metabolism and obesity? Were any known markers of insulin signaling, leptin, etc. tested?

Review form: Reviewer 2

Recommendation

Accept with minor revision (please list in comments)

Do you have any ethical concerns with this paper?

No

Comments to the Author

In this study, Muha and collaborators evaluated the impact of the loss of CRMP2 O-GlcNAcylation on cognitive function in mice. It is a very well design study. Despite the modest results it would be interesting to evaluate the levels of CRMP2 phosphorylation to confirm the crosstalk between these two post-translational modifications and evaluate the potential presence of neurofibrillary tangles (NFTs).

Decision letter (RSOB-19-0192.R0)

07-Oct-2019

Dear Professor van Aalten

We are pleased to inform you that your manuscript RSOB-19-0192 entitled "Loss of CRMP2 O-GlcNAcylation leads to reduced novel object recognition performance in mice" has been accepted by the Editor for publication in Open Biology. The reviewer(s) have recommended publication, but also suggest some minor revisions to your manuscript. Therefore, we invite you to respond to the reviewer(s)' comments and revise your manuscript.

Please submit the revised version of your manuscript within 14 days. If you do not think you will be able to meet this date please let us know immediately and we can extend this deadline for you.

- 1) A text file of the manuscript (doc, txt, rtf or tex), including the references, tables (including captions) and figure captions. Please remove any tracked changes from the text before submission. PDF files are not an accepted format for the "Main Document".
- 2) A separate electronic file of each figure (tiff, EPS or print-quality PDF preferred). The format should be produced directly from original creation package, or original software format. Please note that PowerPoint files are not accepted.
- 3) Electronic supplementary material: this should be contained in a separate file from the main text and meet our ESM criteria (see <http://royalsocietypublishing.org/instructions-authors#question5>). All supplementary materials accompanying an accepted article will be treated as in their final form. They will be published alongside the paper on the journal website and posted on the online figshare repository. Files on figshare will be made available approximately one week before the accompanying article so that the supplementary material can be attributed a unique DOI.

Online supplementary material will also carry the title and description provided during submission, so please ensure these are accurate and informative. Note that the Royal Society will not edit or typeset supplementary material and it will be hosted as provided. Please ensure that the supplementary material includes the paper details (authors, title, journal name, article DOI). Your article DOI will be 10.1098/rsob.2016[last 4 digits of e.g. 10.1098/rsob.20160049].

- 4) A media summary: a short non-technical summary (up to 100 words) of the key findings/importance of your manuscript. Please try to write in simple English, avoid jargon, explain the importance of the topic, outline the main implications and describe why this topic is newsworthy.

Images

Data-Sharing

It is a condition of publication that data supporting your paper are made available. Data should be made available either in the electronic supplementary material or through an appropriate repository. Details of how to access data should be included in your paper. Please see <http://royalsocietypublishing.org/site/authors/policy.xhtml#question6> for more details.

Data accessibility section

Sincerely,

The Open Biology Team

<mailto:openbiology@royalsociety.org>

Reviewer(s)' Comments to Author:

Referee: 1

Comments to the Author(s)

1. By other reports, it is known that O-GlcNAcylation is decreased in AD patients, and as such increased O-GlcNAc is hypothesised to confer protective effects in the context of AD (Tau for example). Can you elaborate on the observation that O-GlcNAcylation of CRMP-2 is not statistically different from normal to AD? Would this data look different if this was performed in a CRMP3 (also hippocampal in mature neurons) knockdown background in a cell line (WT versus AD) that may remove any compensation to tease out the role of the S517 of CRMP2? If most CRMPs are not present in the adult brain, how exactly is this compensation mechanism in play in the 3-6 month old adult mice?
2. Are any of the other CRMPs O-GlcNAcylated?
3. Is there any available information on known SNP variants of CRMP2 that are correlated in any neurodegenerative diseases? Are any of the S/T sites discussed in this report known to segregate with disease?
4. O-GlcNAcylation plays a role in protein-protein interactions. Knowing that CRMP2 and Tau are both involved in neurofibrillary tangles, and associated with each other, do any of the other known O-GlcNAc modified proteins in microtubules interact with CRMP2? Is this interaction lost with the 517A mutant thereby contributing to or causing the effects seen with the phenotypes seen?
5. Considering that the authors are drawing parallel (as inferred from ancillary data) to healthy versus neurodegeneration (in this case AD) of aging human brains, the more appropriate age group for the mouse study should be 14+ months. Is there any data available for mice older than 6 months?
6. Several of the western blot images do not show molecular weight markers in the supplementary figures, Fig. 3G and 1E.

7. Do the authors have an explanation that when tested can elaborate on the weight loss phenotype knowing the relation of O-GlcNAc and OGT to nutrient sensing, glucose metabolism and obesity? Were any known markers of insulin signaling, leptin, etc. tested?

Referee: 2

Comments to the Author(s)

In this study, Muha and collaborators evaluated the impact of the loss of CRMP2 O-GlcNAcylation on cognitive function in mice. It is a very well design study. Despite the modest results it would be interesting to evaluate the levels of CRMP2 phosphorylation to confirm the crosstalk between these two post-translational modifications and evaluate the potential presence of neurofibrillary tangles (NFTs).

Author's Response to Decision Letter for (RSOB-19-0192.R0)

See Appendix A.

Decision letter (RSOB-19-0192.R1)

05-Nov-2019

Dear Professor van Aalten,

We are pleased to inform you that your manuscript entitled "Loss of CRMP2 O-GlcNAcylation leads to reduced novel object recognition performance in mice" has been accepted by the Editor for publication in Open Biology.

Article processing charge

Please note that the article processing charge is immediately payable. A separate email will be sent out shortly to confirm the charge due. The preferred payment method is by credit card; however, other payment options are available.

Sincerely,

The Open Biology Team
mailto: openbiology@royalsociety.org

Appendix A

Dundee, November 4th, 2019

Dear Editor

Thank you for returning the referee comments on our manuscript “Loss of CRMP2 O-GlcNAcylation leads to reduced novel object recognition performance in mice” RSOB-19-0192. We are pleased to see the referees found our manuscript of interest and recommended publication, provided a number of issues were addressed in a revised version. Please find attached a detailed response to their comments. We hope that the manuscript is now acceptable for publication in Open Biology.

Warmest regards,

Prof. Daan van Aalten

Response to the referees' comments

REFEREE 1

We thank the referee for their constructive criticism and suggestions for further work. Some of these are beyond the scope of this work (e.g. the ageing study), but we have addressed them individually below.

“By other reports, it is known that O-GlcNAcylation is decreased in AD patients, and as such increased O-GlcNAc is hypothesised to confer protective effects in the context of AD (Tau for example). Can you elaborate on the observation that O-GlcNAcylation of CRMP-2 is not statistically different from normal to AD? Would this data look different if this was performed in a CRMP3 (also hippocampal in mature neurons) knockdown background in a cell line (WT versus AD) that may remove any compensation to tease out the role of the S517 of CRMP2? If most CRMPs are not present in the adult brain, how exactly is this compensation mechanism in play in the 3-6 month old adult mice?”

The original hypothesis of this work was that O-GlcNAcylation affected phosphorylation of CRMP2 through a competition mechanism. It was with this in mind that we studied levels of O-GlcNAcylation on CRMP2 in AD brains, where phosphorylation had previously been demonstrated to be affected, suggesting that such competition, in the context of the data set used, was not present. We agree with the referee that further studies could be conducted in a genetic background that would remove the possible compensatory mechanisms from other CRMPs, however this is beyond the scope of this study, and we feel this would be risky given that we were not able to demonstrate changes in (CRMP2) phosphorylation upon loss of CRMP2 O-GlcNAcylation.

“Are any of the other CRMPs O-GlcNAcylated?”

There have not been any reports of O-GlcNAcylation of other CRMPs and the site of CRMP2 O-GlcNAcylation is not conserved in the other members of the family. Nevertheless, this does not exclude the possibility of (functional) O-GlcNAcylation on other CRMPs.

“Is there any available information on known SNP variants of CRMP2 that are correlated in any neurodegenerative diseases? Are any of the S/T sites discussed in this report known to segregate with disease?”

We have inspected the gnomAD database (that collates genetic variation in the human population from a variety of sources) showing that there are no missense variants at the Ser517 position. Missense mutations at the phosphorylation sites Thr509 and Ser518 have been reported but at very low frequency (1.19×10^{-5}). The ClinVar database only reports a single pathogenic CRMP2 variant at a position in the catalytic core, far away from the Ser517 region.

“O-GlcNAcylation plays a role in protein-protein interactions. Knowing that CRMP2 and Tau are both involved in neurofibrillary tangles, and associated with each other, do any of the other known O-GlcNAc modified proteins in microtubules interact with CRMP2? Is this interaction lost with the 517A mutant thereby contributing to or causing the effects seen with the phenotypes seen?”

The referee is asking whether the interactome of wild type CRMP2 and S517A CRMP2 is different. This is a really interesting question, but would require a comparison of *stoichiometrically* O-GlcNAcylated CRMP2 with the S517A mutant, in order to perform accurate and meaningful quantitative proteomics. Unfortunately, tools to stoichiometrically incorporate a single O-GlcNAc site within the context of an otherwise unperturbed O-GlcNAc proteome have not been reported yet.

“Considering that the authors are drawing parallel (as inferred from ancillary data) to healthy versus neurodegeneration (in this case AD) of aging human brains, the more appropriate age group for the mouse study should be 14+ months. Is there any data available for mice older than 6 months?”

We agree with the referee that (larger) effects at 14+ month cannot be excluded. However, the initial biochemical dissection reported in the manuscript showed that (at 6 months) there was no evidence to support the original hypothesis that phosphorylation of CRMP2 was affected – and in the context of previous work that would require to perturb the CRMP2 phosphorylation \leftrightarrow AD axis. This was the reason why we then focused on “O-GlcNAc – intrinsic” functions (as opposed to phosphorylation competition) during the first 6 months, giving the weight/memory phenotypes reported in the paper.

“Several of the Western blot images do not show molecular weight markers in the supplementary figures, Fig. 3G and 1E.”

We thank the referee for spotting this and this has now been corrected in the revised manuscript.

“Do the authors have an explanation that when tested can elaborate on the weight loss phenotype knowing the relation of O-GlcNAc and OGT to nutrient sensing, glucose metabolism and obesity? Were any known markers of insulin signaling, leptin, etc. tested?”

The referee points to the large body of work that has shown that O-GlcNAc homeostasis is important for metabolism and the insulin response, as demonstrated with the help of OGA/OGT overexpression/knockdown/knockout studies and OGA inhibitors. We cannot exclude that disruption of the O-GlcNAc site on CRMP2 results in altered neuronal signalling leading to weight phenotypes. However, in this manuscript we have focused on the possible role of Ser517 O-GlcNAcylation in competing with regulatory phosphorylation. Further studies can be directed at dissecting the phosphorylation-independent mechanisms underpinning the phenotypes in the CRMP2 Ser517 O-GlcNAc deficient mouse.

REFEREE 2

We thank this referee for their appreciation of our work – there are no issues to address.